# MUTEX: Learning Unified Policies from Multimodal Task Specifications

**Rutav Shah**[1]
rutavms@cs.utexas.edu

**Roberto Martín Martín**[1*]
robertomm@cs.utexas.edu

**Yuke Zhu**[1*]
yukez@cs.utexas.edu

**Abstract:** Humans use different modalities, such as speech, text, images, videos, etc., to communicate their intent and goals with teammates. For robots to become better assistants, we aim to endow them with the ability to follow instructions and understand tasks specified by their human partners. Most robotic policy learning methods have focused on one single modality of task specification while ignoring the rich cross-modal information. We present MUTEX, a unified approach to policy learning from multimodal task specifications. It trains a transformer-based architecture to facilitate cross-modal reasoning, combining masked modeling and cross-modal matching objectives in a two-stage training procedure. After training, MUTEX can follow a task specification in any of the six learned modalities (video demonstrations, goal images, text goal descriptions, text instructions, speech goal descriptions, and speech instructions) or a combination of them. We systematically evaluate the benefits of MUTEX in a newly designed dataset with 100 tasks in simulation and 50 tasks in the real world, annotated with multiple instances of task specifications in different modalities, and observe improved performance over methods trained specifically for any single modality. More information at https://ut-austin-rpl.github.io/MUTEX/

**Keywords:** Multimodal Learning, Task Specification, Robot Manipulation

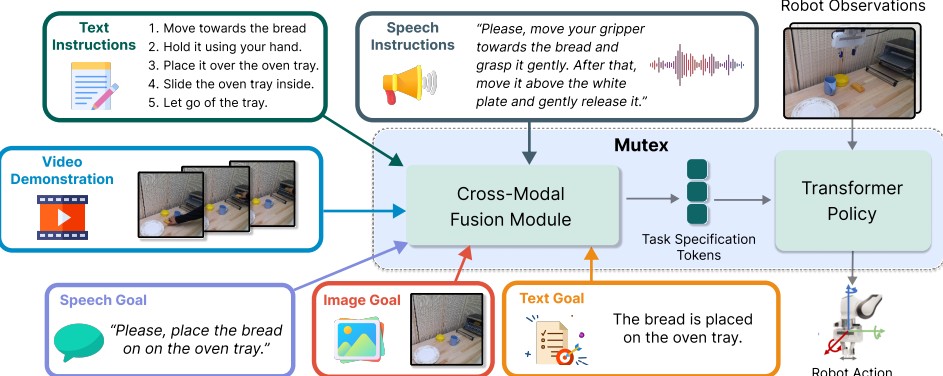

Figure 1: **Overview.** We introduce MUTEX, a unified policy that learns to perform tasks conditioned on task specifications from multiple modalities (image, video, text, and speech) in the forms of instructions and goal descriptions. MUTEX takes advantage of the complementary information across modalities to become more capable of completing tasks specified by any single modality than methods trained specifically for each one.

## 1 Introduction

When working in a team, humans regularly make use of different modalities to specify tasks and improve communications, *e.g.*, sharing high-level task goals ("Let's cook a meal!"), verbal instructions ("We will go to the kitchen, get the pot from the cabinet, and then put it on the stove."), or

---

*Equal advising, [1]The University of Texas at Austin

7th Conference on Robot Learning (CoRL 2023), Atlanta, USA.

fine-grained visual demonstrations (showing a cooking video). Human-robot teams should aspire to a similar level of understanding. While recent research in robot learning has studied various modalities for specifying robotic tasks, including text, images, speech, and videos, most previous studies have treated these individual modalities as separate problems, such as language-conditioned policy learning [1, 2, 3, 4], instruction following [5], visual goal-reaching [6, 7, 8, 9], and imitation from video demonstrations [10, 11, 12]. Consequently, these approaches lead to siloed systems tailored to individual task specification modalities.

A burgeoning body of interdisciplinary AI research has suggested that joint learning across multiple modalities, such as image and text [13, 14, 15, 16], video and text [17, 18, 19] and vision and touch [20, 21, 22], gives rise to richer and more effective representations that improve understanding of individual modalities. These results align with findings in cognitive science and psychology, which suggest that incorporating multimodal cues (*e.g.*, visual and verbal) into human learning processes enhances learning quality over using individual modalities alone [23, 24]. Drawing inspiration from the effectiveness of cross-modal learning in prior work, our goal is to develop **unified policies capable of reasoning about multimodal task specifications for diverse manipulation tasks**, where each task will be defined in a single modality that changes from task to task. We seek to harness the complementary strengths of different modalities — some providing compact and high-level information like goal descriptions, while others providing fine-grained information like step-by-step video demonstrations — thereby improving the model's ability to execute tasks from task specifications of different modalities.

Prior works primarily focus on enhancing language understanding using visual data [25, 26, 27, 28] or on a subset of modalities such as language and robot demonstration [29], language and image [30]. These approaches typically cover one or two modalities, failing to encompass the diverse ways in which humans express their goals and intent. The key challenge associated with learning across varied modalities is effectively leveraging cross-modal information to reinforce each other and having a highly versatile architecture to encapsulate the variability introduced by multiple modalities.

To effectively learn from different task specification modalities, we improve upon two representation learning techniques, **masked modeling** [13, 18, 31] and **cross-modal matching** [14, 32], to foster cross-modal interaction through a shared embedding space. Firstly, we exploit the complementary strengths of each modality — text and speech specifications provide guidance for the model to extract task-relevant features from visual specifications (image goals and video demonstration), and visual specifications, in turn, help ground the text and speech to real-world observations. The model is trained on the representation learning objectives in tandem with the policy learning objective (*i.e.*, behavior cloning) such that the representation of the task specification also captures action-relevant information. After we build richer, more informed representations for each modality, we bring them to a common space [13, 14, 18, 31]. Unlike prior work that maps visual specifications to language embeddings [33], we exploit the fact that human video demonstrations contain more fine-grained information about the task. We enrich the representations of other modalities by matching them with the information-dense video representations. This cross-modal matching leads to compact yet informative multimodal representations that can be used to execute tasks specified by any modality.

For the model to execute a task specified by any modality, it must handle the variable input length of task specification tokens. Meanwhile, it has to predict robot actions alongside a variable number of masked signals for masked modeling of different modalities. To achieve this, we design a Perceiver-style encoder [34] where a variable number of task specification tokens are attended with a fixed history of robot observations using the cross-attention mechanisms. The embeddings obtained from the encoder are then passed through the Perceiver-style decoder [35] to predict robot actions and masked signals for individual modalities. This architecture design allows the model to learn a policy that can execute tasks specified by any single modality or arbitrary aggregation of several.

In summary, we introduce MUTEX (**MU**ltimodal **T**ask specification for robot **EX**ecution), a unified policy capable of executing tasks specified as goal descriptions or more detailed instructions in text, speech, or visual modalities. MUTEX is versatile and performant. It can not only understand multimodal task specifications but also improve the robustness of task execution for every single

modality. We demonstrated this with a comprehensive evaluation benchmark, including 100 diverse manipulation tasks in simulation and 50 in the real world, leading to 6000 evaluated trials (per method) in simulation and 600 evaluation trials in the real world. Remarkably, real-world evaluation indicates that MUTEX can effectively interpret human video demonstrations and perform tasks successfully with the robot, albeit with morphology differences. As part of this effort, we provide a large real-world dataset of 50 tasks for a complete list of real-world tasks with 30 trajectories for each task containing tasks like "putting bread on a microwave tray and closing it", "opening an air fryer and putting hot dogs in it", or "placing the book in the front compartment of the caddy", specified with multiple rich multimodal task specifications: text goal description, text instruction, goal image, video specification, speech goal description, and speech instruction [Figure 3], supporting future research in multimodal task specification.

## 2 Related Work

**Task Specification in Robot Manipulation.** One way to communicate tasks to a multi-task policy is through one-hot encoding vectors [36]. However, this approach is limited to a predefined set of tasks and cannot be extended to new ones. On the other hand, methods that use language to specify tasks [1, 33, 2, 4, 37, 38] have shown improved multi-task generalization due to a richer semantic task representation. However, learning with language specification can be ineffective as it requires grounding language to the robot's observation and action spaces [39, 40]. Moreover, the compact nature of language poses a challenge for tasks that need more detailed or accurate descriptions [30]. Visual task specifications (images [7, 9, 8, 41] or videos [10, 11, 33, 12]) offer dense information which makes the policies falsely depend on task-irrelevant information (*e.g.*, a visual demonstration of moving an object also shows the locations and motions of background objects in the scene [42]), causing poor generalization behaviors. The peculiarity of individual modalities limits methods that focus on unimodal specifications and fail to leverage complementary information across modalities.

**Cross-modal Representation Learning.** In the past years, there has been a large body of literature on learning rich representation from multimodal data with cross-modal learning objectives [16, 43, 44]. These works have provided convincing evidence that learning across multiple modalities can substantially boost model performances on conventional visual recognition tasks (such as image classification [15, 45, 46, 47], object detection [48, 49, 50], segmentation [51, 46], activity recognition [52]), language understanding tasks (such as sentiment analysis, paraphrasing [53, 54]), and multimodal reasoning tasks (such as visual QA [55, 19, 56] and cross-modal retrieval [13, 14]). In robotics, multi-sensory observations have been shown to improve the performance of manipulation tasks [20, 21, 22]. Inspired by these successes, MUTEX learns a cross-modal representation of task specifications for multi-task imitation learning for robot manipulation.

**Multi-Task Imitation Learning in Manipulation:** Multitask imitation learning for robot manipulation has been extensively studied with language task specifications [1, 37, 33, 4] and visual demonstration [12, 9], respectively. A closer line of work to ours consists of methods that harness multimodal task specifications. Some leverage multimodal data for model training, but the final policies are deployed to only operate on a single modality type [25, 26, 27, 33]. Others demonstrate that policies consuming multimodal specifications can generalize to novel task instances in one shot, but the new task must be specified in *all* modalities [29]. More recent works have explored specifying a task with multimodal prompts (text and image tokens) defined in a combination of modalities [30], providing a more flexible interface and partially alleviating grounding problems. Nevertheless, none of these prior works has offered the flexibility to specify the task using *any* individual modality, nor support as many different modalities as MUTEX.

## 3 MUTEX Model and Dataset

Our goal is to learn a unified policy that performs diverse tasks based on a dataset of demonstrations annotated with multimodal task specifications, including language, speech, and visual specifications. We assume that, during test time, the task to perform will be specified in a single or

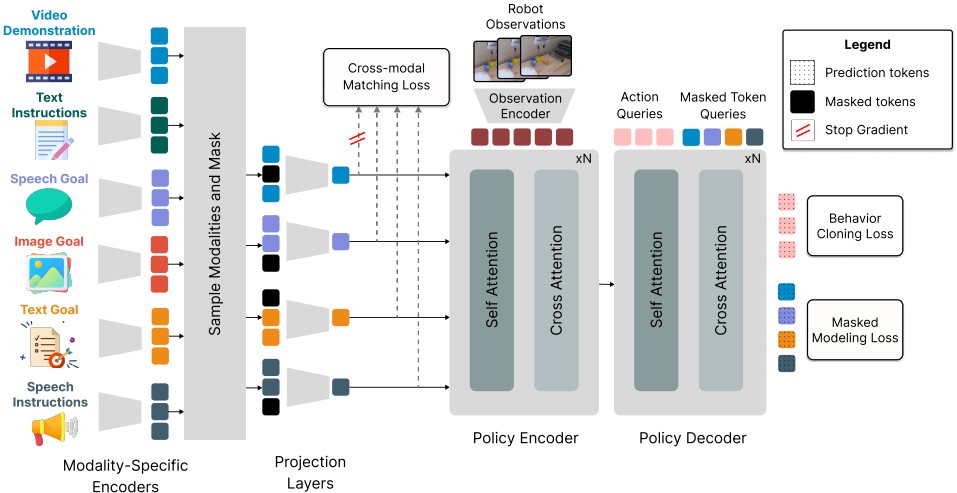

Figure 2: **MUTEX's Model Architecture and Training Losses**. Task specifications in each modality are encoded with pretrained modality-specific encoders, CLIP, and Whisper models [15, 57]. During the first stage of training, one or more of these modalities are randomly selected and masked before being passed to projection layers. The resultant tokens obtained from projection layers are combined with observation tokens through $N$ blocks of self- and cross-attention layers. The encoder's hidden state is passed to MUTEX's transformer decoder that is queried for actions (behavior learning loss) and the masked features and tokens (masked modeling loss), promoting action-specific cross-modal learning. In the second stage of training, all modalities are enriched with information from video features through a cross-modal matching loss. MUTEX predicts closed-loop actions to achieve the task based on the provided observations and a task specification modality (one or more) at test time.

a subset of modalities that can vary from task to task. For each task, $T_i \in \{T_1, T_2, \ldots, T_n\}$, we assume that the agent learns from a set of human demonstrations obtained through teleoperation, $D_i = \{d_i^1, d_i^2, \ldots, d_i^m\}$, where $m$ is the number of demonstrations for task $T_i$, forming an entire dataset of demonstrations, $D$. Each demonstration $d_i^j$ presents the form of a sequence of observations and expert actions, $d_i^j = [(o_1, a_1)_i^j, \ldots (o_T, a_T)_i^j]$.

The goal of each task $T_i$ is specified by $k$ alternative task descriptions ($t$) in six different forms within three modalities: text instructions $t \in \{L_i^1, L_i^2, \ldots, L_i^k\}$, text goal description $t \in \{l_i^1, l_i^2, \ldots, l_i^k\}$, video demonstration $t \in \{V_i^1, V_i^2, \ldots, V_i^k\}$, goal image $t \in \{v_i^1, v_i^2, \ldots, v_i^k\}$, speech instructions $t \in \{S_i^1, S_i^2, \ldots, S_i^k\}$, and speech goal description $t \in \{s_i^1, s_i^2, \ldots, s_i^k\}$. Note that we characterize the modalities with the letters $L/l$, $V/v$, and $S/s$, and make use of capital letters to denote *detailed instructions* and small letters to denote *goal state specifications*. Our goal is to learn a unified policy, $\pi(a|t, o)$, that outputs continuous actions, $a \in A$, given current observations, $o \in O$, conditioned on a task specification in one or more of the possible modalities, $t \in L|l|V|v|S|s$. We aim to create a policy that not only performs the $n$ tasks in the training dataset $D$ (seen tasks) in new initial conditions (*e.g.* positions of the objects) but also generalizes to previously unseen task descriptions.

## 3.1 MUTEX Training Procedure

Our goal is not only to obtain a unified policy that understands task specifications in different modalities but also to improve the policy performance when a task is specified on every single modality by exploiting cross-modal interactions during training. To that end, we leverage two representation learning techniques that we integrate sequentially in MUTEX's training procedure: *1) Mask Modeling* to promote cross-modal interactions of all modalities into a shared learned latent space, and *2) Cross-Modal Matching* to enrich each modality with information of the information-denser one. Both stages of our procedure are combined with a behavior cloning objective for policy learning to ensure that the learned representation contains relevant information for the agent to perform the manipulation task. The overview of the proposed approach is shown in Fig. 2.

**Masked Modeling for Cross-Modal Learning:** In the first stage [Step 1 in PseudoCode 1], MU-TEX promotes cross-modal interactions between the model components interpreting the different

modalities: text instructions ($L$), text goal ($l$), video demonstration ($V$), image goal ($v$), speech instructions ($S$), and speech goal ($s$). Inspired by the success in other cross-modal learning tasks [53, 19, 43, 44], we mask certain tokens or features of each modality and learn to predict them with the help of other modalities. This enforces the model to use the information from other modalities to enhance the representation of each one of them. Intuitively, text- and speech-masked prediction helps in focusing relevant information from visual modalities, while image- and video-masked prediction helps in grounding other modalities. During testing, a task specification in only one or a subset of the modalities is used. Thereby, to obtain robust single-modality representations [Table 1, 2], we recreate these conditions in our training procedure by randomly sampling modalities in each iteration.

Specifically, at each iteration of the training process, we randomly select task specifications of one or a subset of modalities. If more than one modality is selected, we mask certain parts of each modality. We require MUTEX to predict the masked parts alongside the action values the expert demonstrated at each step. Different modalities would mask either input or intermediate features and use a different metric loss to measure prediction error ($\ell_1$ or $\ell_2$). For *masked text modeling* (masking elements of a text goal description or text instructions), we mask out words [58] that are then passed through pretrained CLIP language model [15] to extract the features. We use the standard cross-entropy loss between the predicted ($\hat{y}$) and ground truth ($y$) tokens, *i.e.*, $\mathcal{L}_{CE}(y, \hat{y}) = -\sum_{i=1}^{N} y_i \log(\hat{y}_i)$, where $N$ is the number of tokens in the vocabulary. For *masked visual modeling* (masking elements of an image goal or a video instruction), we mask out intermediate regions of the features obtained from a pre-trained CLIP model [15] and, following prior works in vision-language modeling [59], we employ a simple $\ell_1$-regression loss between the predicted and ground truth features, $\mathcal{L}_{\ell_1} = |y - \hat{y}|$. For *masked speech modeling* (masking elements of a speech goal description or speech instructions), we use a similar approach to visual modeling but use features from a pre-trained Whisper model [57] instead with an $\ell_1$-regression loss (Refer to Appendix 6.2).

**Cross-Modal Matching for Richer Representations:** In the second stage of MUTEX's training procedure [Step 2 in PseudoCode 1], we enrich the common embedding space for each task modality by associating it with the features of the information-richer one. To that end, in contrast to prior works that use a cross-modal contrastive loss to learn a common embedding space [14, 16], we use a simple $\ell_2$ loss to pull the representations of all modalities towards the one with more information and better performance. Video specifications contain the most information, leading to more elucidative and stronger features; therefore, we enrich other modalities with information from the video representation space obtained after cross-modal learning.

Concretely, let $f_L$, $f_l$, $f_v$, $f_s$, $f_S$ be the feature representations of the text instructions, text goal, image goal, speech goal, and speech instructions, and $f_V$ the feature representation of the video demonstration for the same task. Our cross-modal matching loss is given by:

$$\mathcal{L}_{match} = \mathcal{L}_{\ell_2}(f_L, f_V) + \mathcal{L}_{\ell_2}(f_l, f_V) + \mathcal{L}_{\ell_2}(f_S, f_V) + \mathcal{L}_{\ell_2}(f_s, f_V) + \mathcal{L}_{\ell_2}(f_s, f_V) \quad (1)$$

where $\mathcal{L}_{\ell_2}$ is the $\ell_2$-regression loss. Gradients from this loss are not backpropagated to the part of MUTEX that encodes the video modality, leaving them unchanged.

### 3.2 MUTEX Architecture

The training process delineated above requires a model architecture capable of propagating and encoding the cross-modal information from multiple modalities. MUTEX's model consists of three main components (see Fig. 2): 1) Modality-Specific Encoders that map input modalities to task-specific tokens, 2) a Policy Encoder that takes in the task-specific tokens and robot observations and outputs hidden states, and 3) a Policy Decoder that takes in the hidden states along with decoder queries and outputs features corresponding to the queries.

MUTEX's **modality-specific encoders** extract tokens from input task specification using a fixed, pre-trained large model, which helps us to extract semantically meaningful representation from the input modality. To learn representations that are grounded to the observation and action space, these features are passed through a projection layer consisting of a simple MLP or single attention block

before passing it to the policy encoder. MUTEX's **policy encoder** effectively fuses information obtained from multiple task specification modalities and robot observations, employing a transformer-based architecture with stacked cross- and self-attention layers. In the cross-attention layers, queries are derived from robot observations, whereas the keys and values are from task specification tokens. The encoder's output is then passed to the policy decoder. Although the policy encoder's output is enriched with information obtained from different task specification modalities, MUTEX requires the policy to output features for predicting action values and a variable number of masked tokens. This motivates us to adopt a Perceiver Decoder [35] architecture as MUTEX's **policy decoder** to leverage learnable queries and output only the information corresponding to input queries. The decoder features for action prediction are passed through an MLP to estimate a Gaussian Mixture Model for continuous action values. Similarly, separate MLPs are used to predict token values or features for the masked token queries. More details can be found in Appendix 6.3.

### 3.3 Multimodal Task Specification Dataset

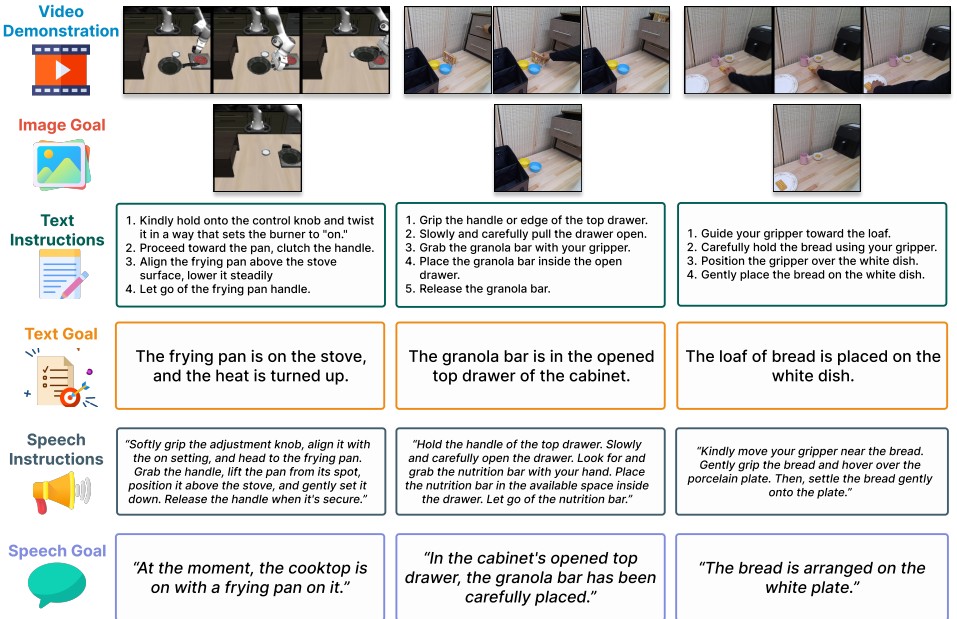

Figure 3: **MUTEX Multimodal Task Specification Dataset.** We provide a dataset comprising **100** simulated tasks (example in the first column) based on LIBERO-100 [60] and **50** real-world tasks (examples in the second and third columns), annotated with **50** and **30** demonstrations per task, respectively. We annotate each task with **11** alternative task specifications in each of the **six** following modalities (rows from top to bottom): video demonstration, image goal, text instructions, text goal, speech instructions, and speech goal.

As part of our efforts to develop a unified multi-task imitation learning policy, we construct a new dataset of tasks with multiple task specifications per modality in both simulated and real-world tasks (see Fig. 3). In the **simulation**, we extend the LIBERO-100 benchmark [60] entailing diverse object interactions and versatile ($n = 100$) tasks like "turn on the stove and put the frying pan on it." Each task is annotated with $m = 50$ human trajectories (provided by the authors) and $k = 11$ task specifications per modality. Due to the inevitable sim2real gap, video demonstrations in the simulation are collected by teleoperating the simulated robot instead of directly by a human performing the tasks. In the **real world**, we collect a novel dataset with $n = 50$ tasks (Figure 5) ranging from pick and place tasks such as "put the bread on the white plate", "pick up the bowl at the back of the scene and place it inside the top drawer" to more contact-rich tasks such as "open the air fryer and put the bowl with dogs in it", "take out the tray of the oven, and put the bread on it." All tasks are collected in the same environment but involve different objects from a set of 17 objects. Each task is demonstrated with $m = 30$ human-collected trajectories [61] using a 3D spacemouse teleoperation device. Each task is annotated with $k = 11$ different task specifications

| Method | Text Goal | Text Instructions | Image Goal | Video Demonstration | Speech Goal | Speech Instructions |
|---|---|---|---|---|---|---|
| Modality-Specific | $41.7 \pm 8.0$ | $39.9 \pm 2.8$ | $58.7 \pm 4.9$ | $62.0 \pm 5.2$ | $22.3 \pm 1.2$ | $28.4 \pm 5.5$ |
| MUTEX (joint training) | $39.2 \pm 9.1$ | $38.3 \pm 7.4$ | $48.6 \pm 14.8$ | $50.7 \pm 16.0$ | $32.1 \pm 9.9$ | $38.2 \pm 13.6$ |
| MUTEX (no masked modeling) | $34.8 \pm 6.0$ | $38.7 \pm 6.5$ | $43.8 \pm 6.0$ | $46.0 \pm 8.0$ | $24.6 \pm 4.2$ | $29.1 \pm 6.1$ |
| MUTEX (no cross-modal matching) | $43.5 \pm 8.9$ | $39.4 \pm 2.2$ | $60.1 \pm 6.4$ | $\mathbf{63.2 \pm 6.3}$ | $36.7 \pm 4.3$ | $\mathbf{46.8 \pm 5.5}$ |
| MUTEX | $\mathbf{50.1 \pm 7.8}$ | $\mathbf{53.0 \pm 2.2}$ | $\mathbf{61.6 \pm 6.4}$ | $\mathbf{63.2 \pm 6.3}$ | $\mathbf{40.9 \pm 8.1}$ | $46.0 \pm 5.2$ |

Table 1: Success Rate on the Multimodal Task Specification Dataset in Simulation.

| Method | Text Goal | Text Instructions | Image Goal | Video Demonstration | Speech Goal | Speech Instructions |
|---|---|---|---|---|---|---|
| Modality-Specific | 52 | 48 | 42 | 52 | 32 | 46 |
| MUTEX | **64** | **58** | **62** | **64** | **50** | **60** |

Table 2: Success Rate on the Multimodal Task Specification Dataset in Real-World.

for each modality. A human performs video demonstrations specifying the tasks with their hand. To generate 11 diverse text goal descriptions and instructions, we make use of ChatGPT with prompts to generate alternative descriptions that we manually filter to avoid using synonyms that do not match the right task-relevant objects (*e.g.*, using *plate* as a synonym for *bowl* when there are other plates in the scene). We request speech signals from several characters of the Amazon Polly service to generate diverse speech descriptions.

## 4 Experimental Evaluation

**Experimental Setup:** We conduct our evaluations on our newly constructed dataset of multimodal task specifications, with a split of 80%/20% (8/3) tasks for training and testing. For evaluation, we subject the robot to both unseen task specifications (from the test split) and new initial conditions (i.e., positions of objects). For each tested task, we evaluated 20 trials in simulation and 10 trials in the real world. Reported results for each evaluation modality are average for all 20 trials $\times$ 100 tasks $\times$ 3 seeds in the simulation and 10 trials $\times$ 5 tasks in the real world. In our evaluation, we compare MUTEX to models trained specifically for each modality using only task specifications in that modality. This resembles most existing prior works in video-based imitation learning [11, 12], and goal-conditioned imitation learning [6, 9, 33, 4]. Please refer to Appendix 6.5 for details.

**Experiments:** In our experimental evaluation, we aim to answer the following questions:

*1) Does a unified policy capable of executing tasks across multiple modalities outperform methods trained specifically from and for each individual modality?* Table 1 and Table 2 summarize the results of our evaluation in simulation and the real world, respectively. In both cases, we observe a significant improvement ($+\mathbf{10.3}\%$, $+\mathbf{14.3}\%$ in simulation and real-world respectively) from using our unified policy MUTEX compared to modality-specific models, indicating that the cross-modal learning procedure from MUTEX is able to leverage more information from other modalities.

Additionally, we analyze the errors in our real-world evaluation and find that representations learned using MUTEX generalize better to new task specifications than unimodal representations. Specifically, when dealing with unseen task specifications, failed trials that were not attributed to BC compounding error, where the policy correctly completed a semantically meaningful task in the environment, albeit not the intended task (e.g., picking and placing another object). We find that in the unimodal baselines, **85** out of **240** trials ($35.4\%$) are due to failed understanding of tasks specified, whereas it reduces to **40** out of **240** ($16.7\%$) with MUTEX.

*2) What is the importance of the two stages of the MUTEX training procedure? Is it important to perform the stages consecutively?* Table 1 includes a comparison of the results obtained with the two-staged training procedure consecutively (MUTEX) compared to training with both stages simultaneously (joint training), training without the first stage (no masked modeling) or without the second stage (no cross-modal matching). We observe the largest drop in performance comes from training without masked modeling, indicating that this step is critical to learning cross-modal

information from different task specifications. Cross-modal matching provides a boost except for video demonstrations (we match to that modality) and observes a small drop in speech specifications.

*3) Does the performance increase significantly when tasks are specified with multiple modalities?* One of the advantages of MUTEX is that it can execute tasks with a single specification in any of the learned modalities or with multiple specifications in several of the modalities. We evaluated using combinations of *Text Goal + Speech Goal*, *Text Goal + Image Goal* and *Speech Instructions + Video Demonstration* and obtain $50.1$, $59.2$, and $59.6$ success rates, respectively. These values are close to the performance using one single specification in the best of the two modalities, indicating that the additional modality is not providing any extra information. We hypothesize it is because all possible cross-modal information has already been learned by MUTEX. Interestingly, when using specifications in *all modalities*, the success rate is $60.1$, lower than when using *Image goal* or *Video Demonstration*, possibly due to the increased complexity of interpreting multiple task specifications.

*4) Are the task specification representations learned by MUTEX better than state-of-the-art task-specification models?* To further evaluate the efficacy of the MUTEX representations, we compare

| Method | Text Goal | Text Instruction | Image Goal | Video Demonstration | Image Goal + Text Goal |
|---|---|---|---|---|---|
| T5 [62] | 40.0 | 44.0 | - | - | - |
| R3M [63] | - | - | 59.5 | 44.7 | - |
| VIMA [30] | - | - | - | - | 47.0 |
| MUTEX | **50.1** | **53.0** | **61.6** | **63.2** | **59.2** |

Table 3: Success Rate on the Multimodal Task Specification Dataset in Simulation.

it with other task specification models in Table 3. Although the unimodal models, T5 and R3M, achieve better results than CLIP (Table 1) in Text Instructions ($+4.1\%$) and Image Goals ($+2\%$), respectively, MUTEX consistently outperforms these models across all modalities. The consistent improvement across modalities demonstrates the value of leveraging multiple modalities during training. Moreover, MUTEX significantly outperforms VIMA, a recent method that employs both text goals and object images for task specification, highlighting that MUTEX is not only more performant than its unimodal counterparts but also can effectively use multiple modalities during inference.

## 5 Conclusion, Limitations, and Future Work

We demonstrate with comprehensive experiments in simulation and the real world that multi-task learning, when trained on task specifications across multiple modalities, produces a more robust and versatile policy in each modality. We are highly encouraged by the empirical results and the potential of MUTEX for designing a more versatile multimodal human-robot communication interface.

We aim to improve MUTEX in future work by addressing several limitations. MUTEX assumes paired access to all the task specification modalities, which may be difficult to obtain in a scalable fashion. MUTEX uses of clean speech signals synthesized by Amazon Polly may not accurately represent real-world speech, which is often noisier and more difficult to understand. MUTEX uses video and image goals are provided from the same workspace as the task to be executed. Having a policy that can execute task specified "in the wild" visual goal or demonstration will invite additional challenges. We also plan to explore how to foster stronger generalization by training across diverse environments, which could open the door to the use of larger human video datasets. Lastly, MUTEX uses vanilla behavior cloning to learn policies, which possesses problems like covariate shifts and compounding errors. To mitigate this limitation, incorporating interactive imitation learning and reinforcement learning techniques is an exciting direction for future research.

**Acknowledgments**

We thank Yifeng Zhu and Huihan Liu for real robot system infrastructure development. We thank Zhenyu Jiang, Jake Grigsby, and Hanwen Jiang for providing helpful feedback for this manuscript. We acknowledge the support of the National Science Foundation (1955523, 2145283), the Office of Naval Research (N00014-22-1-2204), UT Good Systems, and the Machine Learning Laboratory.

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

# 6 Appendix

## 6.1 Model Training Ablations

| Method | Text Goal | Text Instructions | Image Goal | Video Demonstration | Speech Goal | Speech Instructions |
|---|---|---|---|---|---|---|
| MUTEX (w/ pretraining representations) | 42.3 | 46.8 | 46.0 | 48.0 | 34.4 | 32.0 |
| MUTEX (w/ finetuning full model) | 37.1 | 40.2 | 44.7 | 46.3 | 31.6 | 34.2 |
| MUTEX (w/ finetuning video encoder) | 49.6 | **53.4** | 57.6 | 61.3 | 38.4 | 46.0 |
| MUTEX | **50.1** | 53.0 | **61.6** | **63.2** | **40.9** | **46.0** |

Table 4: Model Training Ablations of MUTEX variants.

In MUTEX training, the task specification representations are learned in tandem with the policy learning objective, i.e., behavior cloning. The policy learning objective helps capture the necessary information for robot action prediction, missing in the task-specification dataset. To verify this empirically, we pre-train the task specification representations using the two-stage training process without utilizing the BC loss. Subsequently, we learned the policy using the frozen task specification representations (Referred as **MUTEX (w/ pretraining representations)** in Table 4). MUTEX significantly performs better (**+10.9%**) than MUTEX (w/ pretraining representations).

Cross-modal matching in MUTEX aims to enrich other modalities with the more-informed video specification obtained after masked modeling. To avoid local optima, where the video specification loses information to match representations of other modalities, we freeze the MUTEX Policy and Video Encoder layers. To validate the hypothesis, we evaluate the performance of two variants of MUTEX with finetuning full model (including video encoder)- **MUTEX (w/ finetuning full model)** and finetuning video encoder- **MUTEX (w/ finetuning video encoder)**. The results are highlighted in Table 4. We observe that the training strategy adopted by MUTEX consistently outperforms other variants, corroborating our initial hypothesis that backpropagating gradients through video encoder in cross-modal matching may lead to sub-optimal performance.

## 6.2 Training Details

For training MUTEX 3.2, following the standard practice, we use the AdamW optimizer [64] along with a Cosine Annealing LR Scheduler [65]. To facilitate learning a robust model, we apply data augmentation like ColorJitter and translation augmentation on the RGB observation images. The hyperparameter details are adopted from LIBERO's transformer baseline [60], and the details are provided in Table 5. For masked modeling, we train MUTEX for 50 epochs end-to-end, whereas for the cross-modal matching, the projection layers [Figure 2] are finetuned for 20 epochs. The model is trained using the 2 NVIDIA RTX A5000 (24 GB) GPUs with a batch size of 64 using the PyTorch framework [66] for all the baselines, and MUTEX with an approximate total of five days for each experiment.

**Masked Modeling**: We utilize masked modeling to facilitate intermodal interaction between different modalities [Section 3.1]. For the **text** modality, we mask out specific words from the input and predict the corresponding word tokens. Regarding text goals, we randomly select one word from the

input specification. For text instructions, we repetitively mask out two random words. It is worth noting that stop-words (e.g., 'a', 'the') are not masked, as predicting stop-words does not provide helpful task-related information. Regarding **visual** specifications, we mask out features extracted from CLIP instead of directly predicting noisy pixel values. We employ L1 regression loss for training purposes. We mask the intermediate region feature obtained after the 22nd transformer block in CLIP for image goals. For video demonstrations, we mask out individual frame features. Similarly, for **speech** specifications, we mask and predict features directly obtained from the pretrained Whisper-Small encoder model for both speech goals and speech instructions.

| Training Details | |
|---|---|
| Batch Size | 64 |
| Gradient Clipping | 100 |
| Epochs (Masked Modeling) | 50 |
| Epochs (Cross-Modal Matching) | 20 |
| Initial Learning Rate | 1e-4 |
| Betas | (0.9, 0.999) |
| Weight Decay | 1e-4 |
| Minimum Learning Rate | 1e-5 |
| Masked Modeling Loss Weight | 0.5 |
| Cross-Modal Matching Loss Weight | 0.5 |
| Data Augmentation | |
| Brightness | 0.3 |
| Contrast | 0.3 |
| Saturation | 0.3 |
| Hue | 0.3 |
| Translation | 8 pixels |

Table 5: Hyperparameter details for training MUTEX

## 6.3 Model Architecture

MUTEX leverages multiple modalities during training to effectively learn rich, well-informed representations of each modality, allowing to specify the task using single or any combination of task specification modalities during evaluation. To allow such flexibility, we adopt transformer architecture (Details in main text 3.2).

**Modality Specific Encoders** extract a single embedding for each input task specification modality. To extract semantically meaningful, grounded representations, each modality is passed through a **pre-trained large model**, and then through a **projection layer** [Figure 2]. For (a) *pre-trained large model*, MUTEX uses the CLIP Large ViT-L/14 [15] model for text and visual modalities whereas Whisper-Small [57] encoder for speech modalities providing semantically meaningful embeddings. The (b) *projection layer* layers for each modality enable MUTEX to learn a grounded, common embedding for all modalities. For modalities with single embeddings from the pre-trained model, we use an MLP projection layer [Refer 1], whereas for modalities with multiple embeddings from the pre-trained model, we use a single transformer block with mean pooling [Refer 2] to aggregate information into one embedding. The details of the modality-specific encoders are summarized in Table 6.

Listing 1: Model Architecture for the MLP in Projection Layers

```
MLPBlock(
    nn.Linear(768, 512),
    nn.ReLU()
    nn.Dropout(0.1)
    nn.Linear(512, 768)
    nn.Dropout(0.1)
)
```

| Modality | Pretrained Large Model | Pretrained Embedding Shape | Projection Layer |
|---|---|---|---|
| Text Goal | CLIP | (1, 768) | MLPBlock[1] + Cross-modal Matching MLPBlock[1] + Shared MLPBlock[1] |
| Text Instructions | CLIP | (L, 768) L = num_instructions | TransformerPoolBlock[2] + Cross-modal Matching MLPBlock[1] + Shared MLPBlock[1] |
| Image Goal | CLIP | (1, 768) | MLPBlock[1] + Cross-modal Matching MLPBlock[1] + Shared MLPBlock[1] |
| Video Demonstration | CLIP | (16, 768) | TransformerPoolBlock[2] + Shared MLPBlock[1] |
| Speech Goal | Whisper | (4, 768) | TransformerPoolBlock[2] + Cross-modal Matching MLPBlock[1] + Shared MLPBlock[1] |
| Speech Instructions | Whisper | (4, 768) | TransformerPoolBlock[2] + Cross-modal Matching MLPBlock[1] + Shared MLPBlock[1] |

Table 6: MUTEX's Modality Specific Encoder consists of a pretrained large model and a projection layer for each modality. The embeddings extracted from the pretrained model are transformed into single embeddings using the projection layers. The projection layers consist of a modality-specific MLP or Transformer block, an MLP block added in the cross-modal matching, and a shared MLP block between all modalities.

Listing 2: Model Architecture for the Transformer Block with Mean Pooling in Projection Layers

```
TransformerPoolBlock(
    nn.LayerNorm(768),
    nn.MultiheadAttention(
            embed_dim = 768,
            num_heads = 4,
            kdim = 256,
            vdim = 256
            dropout = 0.1,
    )
    nn.LayerNorm(768),
    TransformerFeedForwardNN(
            nn.Linear(768, 256),
            nn.GELU(),
            nn.Dropout(0.1),
            nn.Linear(256, 768),
            nn.Dropout(0.1)
    )
    nn.AvgPool1d(kernel_size = num_embeddings)
)
```

**Policy Encoder** combines the **robot observations** with the **task specification** embeddings received from the modality specific encoders. The robot observations consist of a history ($T = 10$) of RGB Images from two camera viewpoints (*agentview*, *eye_in_hand*) and proprioceptive information. The RGB images are converted to token embeddings using a ResNet encoder [67] whereas the proprioceptive information is encoded using an MLP [Refer to [60] for more details]. The robot observation embeddings and task specification embeddings are combined using stacked cross-attention and self-attention layers as shown in Figure 4, and the hyperparameters are summarized in Table 7.

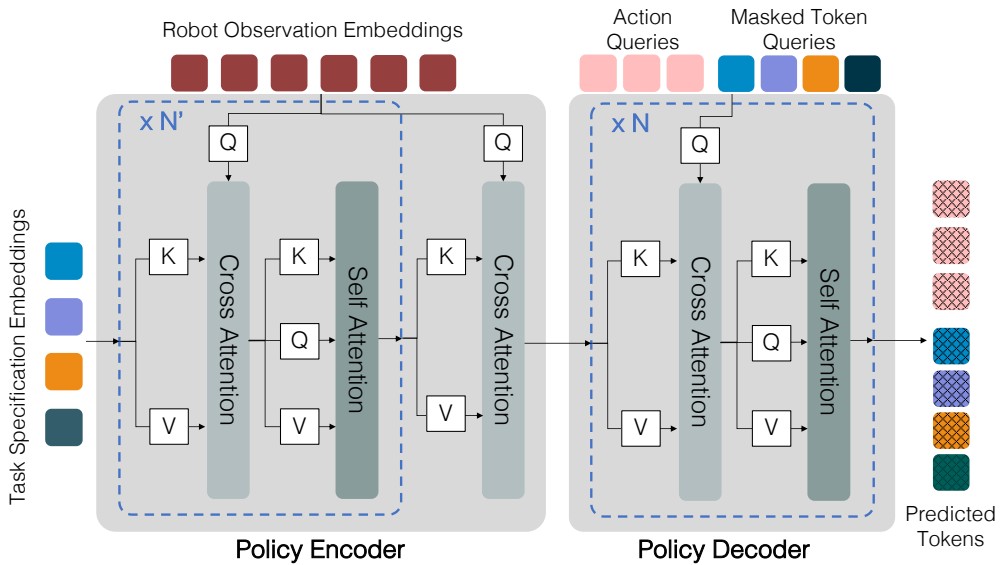

Figure 4: MUTEX Policy Encoder and Decoder Architecture

| Policy Encoder | |
| --- | --- |
| Input Embedding Size | 768 |
| Input Query Embedding Size | 64 |
| Transformer Head Size | 64 |
| # Transformer Heads | 6 |
| Transformer Output Size | 64 |
| Transformer MLP Ratio | 4.0 |
| Dropout | 0.1 |
| Attention Dropout | 0.1 |
| # Self-Attention Layers | 3 |
| # Cross-Attention Layers | 4 |
| Policy Decoder | |
| Input Embedding Size | 64 |
| Input Query Embedding Size | 64 |
| Transformer Head Size | 16 |
| # Transformer Heads | 4 |
| Transformer Output Size | 64 |
| Transformer MLP Ratio | 4.0 |
| Dropout | 0.1 |
| Attention Dropout | 0.1 |
| # Self-Attention Layers | 4 |
| # Cross-Attention Layers | 4 |

Table 7: Hyperparameter details of MUTEX's Policy Encoder and Decoder architecture.

**Policy Decoder** takes in the input from the policy encoder and the learnable queries, enabling MU-TEX to output a variable number of tokens. These outputs are used to predict the masked token values during masked modeling and action values. The learnable query vectors indicate the type of corresponding output - {Action, Text Goal, Text Instructions, Image Goal, Video Demonstration, Speech Goal, Speech Instructions}. These vectors are added with sinusoidal positional vectors indicating the action index and position of the masked tokens in action queries and masked modeling queries, respectively. Similar to policy encoder, it consists of stacked cross-attention and self-attention layers as shown in Figure 4 and the hyperparameters are summarized in Table 7.

**Prediction Heads** [Not shown in Figure 2 for simplicity] using the output of the policy decoder is used to predict the masked token values during masked modeling and action values. For text modeling, the embeddings are mapped to the vocabulary size using a two-layered MLP and trained using cross-entropy loss. For visual and speech modeling, intermediate masked features are predicted using a single-layer MLP and trained using L1 loss. For action modeling, the action values are predicted using a two-layered MLP. The action distribution is modeled as a Gaussian Mixture Model with five components to incorporate the multimodality in human-collected demonstrations and trained using negative log-likelihood loss.

### 6.4 Multimodal Dataset Details

MUTEX **Robot Data**: MUTEX uses behavior cloning to learn from expert datasets collected by humans using teleoperation. We benchmark on two environments: simulation with 100 tasks and real-world manipulation tasks [Refer to 3.3]. For **simulation**, we use LIBERO 100 [60] manipulation benchmarking suite consisting of tasks better representative of the real world. We use the dataset provided by the original authors, which consists of 50 trajectories per task collected using SpaceMouse. For **real world** evaluation, we designed 50 tasks in the real world consisting of pick and place tasks like "Put the bread on the white plate" to more complex tasks like "Open the air fryer and put the bowl with hot dogs in it". The 50 tasks are divided across eight different scenes, with an average horizon length of 242 for each task. A complete list of tasks, along with scene visualization, is provided in Figure 5. We use a Franka Panda robot with a Kinect Azure workspace camera and an Intel RealSense camera for eye-in-hand view. We use SpaceMouse to collect 30 trajectories per task, allowing us to collect 1,500 trajectories. The dataset is publicly available at https://ut-austin-rpl.github.io/MUTEX/, and we hope that it will serve as a resource for future research in real-world manipulation tasks. The real robot infrastructure is adopted from [61].

Listing 3: GPT4 Prompts for generating the annotations of the task

```
# GPT4 Prompt for generating single text specification
# To generate example_paragraph for text instructions
"I will give you instructions; you should break it down into smaller instructions
    for a robot. Please do it without adding any extra objects in the environments
    in the instructions. Break it down into 3-4 instructions, for example, 'pick up
     the apple': 'Move your gripper towards the apply. Hold the apple gently with
    your gripper. Lift the apple from the surface.' Some points to consider while
    generating instructions: The robot can only grasp one object at a time. All the
     objects are in front of the robot, and the robot does not need to move.
Give me one such example for every instruction. Each example must complete the task.
     Here is the task instruction {task_instruction}"

# Prompt for generating multiple text specifications using the afore-generated
    single text specification
"Replace the words with synonyms or rephrase the sentence. Use words that an English
    -speaking person will use in day-to-day routine and simple English. You can
    also add words to make it more polite, such as please, carefully, with caution,
     etc. Give me ten such examples of rephrased paragraphs. Here is the paragraph
    to rephrase: {example_paragraph}"

# Prompt for generating speech annotations using text annotations
"You are a person talking to a robot. Rewrite each paragraph with simple English,
    natural, and polite. Make sure the input is an assertive sentence; keep it
    assertive. If the input is imperative, keep it imperative. Here are {
    text_annotations} such paragraphs, rewrite each paragraph separately:\n"
```

MUTEX **Task Specification Annotations**: MUTEX leverages multiple modalities of specifying the task, including text goal, text instructions, video demonstration, image goal, speech goal, and speech instructions. During training, we assume access to all modalities of task specifications for each task, where each modality helps reinforce other modalities of specification. Thus, we can specify the task using any individual modality during evaluation more effectively. We annotate with 11 task specifications per modality per task [Refer to Figure 3 for examples of task specifications]. We follow the

| Speaker Code | Speaker Name |
|:---:|:---:|
| **Training Speakers** | |
| en-IN | Aditi, Raveena |
| en-AU | Nicole, Russell |
| en-US | Joanna, Matthew |
| en-GB | Amy, Brian |
| **Evaluation Speakers** | |
| en-US | Salli |
| en-GB | Emma |

Table 8: Amazon Polly Codes for the speakers in the speech annotations.

standard protocol of 80%/20% train/eval split for the task specification annotations. All the results are reported on the unseen task specification evaluation set. For simulation, to avoid the real2sim gap, we collected the video demonstration using the teleoperation of the robot in simulation [only RGB information is used to specify the task]. For real-world tasks, human demonstrations are used to specify the task, allowing a more intuitive way of specifying the task for users than teleoperation. For text specifications, we leverage GPT4 to generate initial task specifications for different tasks with diverse variations. The generated task specifications are corrected by humans to ensure the quality of task specification annotations. For speech specifications, we follow a similar procedure to text annotations and, finally, use the Amazon Polly service to generate speech specifications. The prompt used to create annotations for text and speech specifications is provided in Appendix 3, and the details of Amazon Polly characters are provided in Table 8.

## 6.5 Baseline Details

We compare MUTEX against modality-specific baselines for each task specification modality, i.e., Text Goals, Text Instructions, Image Goals, Video Demonstrations, Speech Goals, and Speech Instructions. To highlight the significance of multimodal training proposed in our work, we do not change the policy architecture [Refer 6.3], BC training hyper-parameters [Refer 6.2] in baselines. Moreover, to avoid experimental bias due to model size, we iterate over three different model sizes for each modality-specific baseline with increasing self-attention and cross-attention layers and use the model size with the best performance for the respective modality baseline. The ablated baselines are highlighted in Table 9. The **Modality-Specific** baselines in Table 1 use the same pretrained model as that of corresponding modality in MUTEX, i.e., CLIP [15] for Text Goals, Text Instructions, Image Goals, Video Demonstrations and WHISPER [57] for Speech Goals, Speech Instructions. In Table 3, for **T5 (w/ Text Goals)** and **T5 (w/ Text Instructions)**, we use T5-small [62] model to encode the text specifications similar to language conditioned imitation learning works [2, 33]. Similarly, we use R3M features [63] in **R3M (w/ Image Goals)** and **R3M (w/ Video Demonstratoins)**. We also compare with **VIMA** [30], which uses object images and text goals to specify the task. For the VIMA baseline, we append the text goal features with the top six object crop (pad with zeros otherwise) locations obtained from Image Goal using Mask-RCNN [68]. We observe that one single model, MUTEX, consistently outperforms all the baselines in each evaluation modality. This demonstrates that MUTEX training on multimodal task specifications performs better than training with individual or a subset of modalities.

| Modality Specific Baseline | | | | | | |
|---|---|---|---|---|---|---|
| Model size | Text Goal | Text Instructions | Image Goal | Video Demonstration | Speech Goal | Speech Instructions |
| S (13M) | **47** | **47** | 54 | 66 | **34** | **32** |
| M (14M) | 30 | 30 | **62** | **67** | 20 | 25 |
| L (15M) | 46 | 33 | 56 | 61 | 14 | 21 |

Table 9: Model size ablation for baselines: We iterate over three different model sizes for each modality-specific baseline to avoid experimental bias in model size selection and report results using the best one in Table 1. Note that the results are obtained using one seed and averaged over 100 tasks × 20 trials.

---

**Algorithm 1** PseudoCode for MUTEX training.

---

**input:** Training data: expert robot trajectories, task specifications $t_i = \{L_i, l_i, V_i, v_i, S_i, s_i\}$ for each task $T_i$, where

$$L : \text{Text Instructions}$$
$$l : \text{Text Goals}$$
$$V : \text{Video Demonstrations}$$
$$v : \text{Image Goals}$$
$$S : \text{Speech Instructions}$$
$$s : \text{Speech Goals}$$
$$\theta : \text{Policy Weights}$$
$$\theta_x : \text{Projection layer weights of modality } x \in \{L, l, V, v, S, s\}; \theta_x \subset \theta$$

**output:** Policy network with updated weights

**Step 1:**
**for** $epoch = 1$ *to* $n$ **do**
    # Observation ($o$), Expert Actions ($a_e$), Task Specifications ($t \in \{L, l, V, v, S, s\}$)
    **for** *each batch* $(o, a_e, t)$ **do**
        Sample $k$ task specification modalities: $t' \subseteq t$
        $t' = \text{apply\_masking}(t')$ if $k > 1$
        # Predict: (action, masked tokens)
        $\hat{a}, \hat{t} = \pi_\theta(o, t')$
        # Calculate losses and update policy weights $\theta$
        $\mathcal{L}_{BC} = -\sum_i a_{e,i} \log(\hat{a}_i)$
        $\mathcal{L}_{\text{masked\_modelling}} = L(\hat{t}, t)$ if $k > 1$ $\begin{cases} \text{L1 regression loss for } \{V, v, S, s\} \\ \text{cross entropy loss for } \{L, l\} \end{cases}$
    **end**
**end**

# Freeze all the policy weights except for the projection layer [Refer figure 2] of $L, l, v, S, s$

**Step 2:**
**for** $epoch = 1$ *to* $n'$ **do**
    # Observation ($o$), Expert Actions ($a_e$), Task Specifications ($t \in \{L, l, V, v, S, s\}$)
    **for** *each batch* $(o, a_e, t)$ **do**
        Sample $k$ task specification modalities: $t' \subseteq t$
        # Predict: (action)
        $\hat{a} = \pi_\theta(o, t')$
        # Calculate losses and update policy weights $\theta_L, \theta_l, \theta_v, \theta_S, \theta_s$
        $\mathcal{L}_{BC} = -\sum_i a_{e,i} \log(\hat{a}_i)$
        $\mathcal{L}_{\text{crossmodal\_matching}} = \mathcal{L}_1\big(\pi_{\theta_V}(V), \pi_{\theta_{t'}}(t')\big)$
    **end**
**end**

| Scene Visualization | Task Description | Average Horizon Length |
|---|---|---|
|  | open the bottom drawer | 171 |
| | open the top drawer | 180 |
| | open the top drawer and put the bowl in the back of the scene inside it | 415 |
| | open the top drawer and put the granola bar inside it | 426 |
| | put the granola bar in the back compartment of the caddy | 185 |
| | put the granola bar in the front compartment of the caddy | 197 |
|  | pull out the tray of the oven and put the blue cup on it | 398 |
| | pull out the tray of the oven and put the red bowl on it | 398 |
| | put the blue cup in the basket | 143 |
| | put the mac and cheese box in the basket | 129 |
| | pull out the tray of the oven | 188 |
| | put the red bowl in the basket | 143 |
|  | put the granola bar inside the basket | 135 |
| | put the granola bar inside the top drawer | 163 |
| | put the pink mug inside the basket | 176 |
| | put the red bowl inside the basket | 150 |
| | put the pink mug inside the top drawer and close the drawer | 375 |
| | put the red bowl inside the top drawer and close the drawer | 410 |
| | close the top drawer and open the bottom drawer | 277 |
|  | pull out the tray of the oven | 182 |
| | pull out the tray of the oven and put the bowl with hot dogs on the tray | 436 |
| | pull out the tray of the oven and put the red cup on the tray | 403 |
| | put the book in the back compartment of the caddy | 193 |
| | put the book in the front compartment of the caddy | 199 |
| | put the red cup in the back compartment of the caddy | 161 |
|  | push the tray of the oven in | 174 |
| | put the blue mug on the oven tray | 167 |
| | put the blue mug on the white plate | 146 |
| | put the bread on oven tray and push it in the oven | 429 |
| | put the bread on the white plate | 148 |
| | put the yellow bowl on oven tray and push it in the oven | 418 |
| | put the yellow bowl on the white plate | 143 |
|  | open the air fryer | 208 |
| | open the air fryer and put the bowl with hot dogs in it | 410 |
| | open the air fryer and put the bread in it | 426 |
| | put the bowl with hot dogs on the white plate | 155 |
| | put the bread on the white plate | 145 |
| | put the pink mug on the white plate | 156 |
|  | open the air fryer and put the blue bowl inside it | 388 |
| | open the fryer | 182 |
| | put the book in the back compartment of the caddy | 194 |
| | put the book in the front compartment of the caddy | 193 |
| | put the red cup in the front compartment of the caddy | 158 |
| | put the blue bowl in the back compartment of the caddy | 146 |
|  | open the bottom drawer | 157 |
| | open the top drawer | 172 |
| | open the top drawer and put the blue mug inside it | 404 |
| | open the top drawer and put the pink mug inside it | 432 |
| | put the blue mug on the white plate | 153 |
| | put the bread on the white plate | 158 |
| | Total number of tasks = 50 | 241.9 |

Figure 5: MUTEX real-world task visualization: MUTEX real-world tasks consist of diverse tasks like "opening the top drawer", "placing the bread on oven tray", "putting a bowl consisting of hot dogs inside an air fryer after opening it" representing the complexity of tasks in a real home-kitchen environment.

