# OpenReview forum: "MUTEX: Learning Unified Policies from Multimodal Task Specifications"
_robot-learning.org/CoRL/2023/Conference — CoRL 2023 Poster_

### Official Review · Reviewer_n91Z · 2023-06-24

**Confidence:** 3
**Originality:** Good
**Technical Quality:** Good
**Clarity Of Presentation:** Good
**Impact:** 3

**Recommendation:**

Weak Accept: I recommend accepting the paper, but will not argue for my recommendation if the majority of other reviewers have a different opinion.

**Review:**

Strength:

1. The motivation is sound that as humans, we can specify tasks with different methods and can generalize well. Recent vision results such as segment-anything also show the importance of considering multiple ways of prompting to learn a generalizable representation space.
2. The evaluation is thorough, 100s of tasks with 1000s of trials each.
3. The cross-masked modeling procedure and the transformer architecture (with pretrained encoders for speech and language instructions) makes sense and seemed to work well.

Weakness:
1. The current method relies on paired labels for each modality, which is expensive and prohibitive. Would be interested to see if the framework can still work with in-the-wild human demonstration data, or larger train-test gaps.
2. The overall results seem to indicate a small improvement over only video-conditioned policy, which is, from my understanding, trained from scratch and less practical for test-time.
3. The overall writing and figures an be improved.


**Quality Of The Limitations Section:**

Additional details required

**Questions For Rebuttal:**

Based on the discussions I assume the policy is a multitask policy with 80/20 split on the task-level, and most of the primitives are pick and place? What about human tasks? Are they used in joint training? Are there any sim-to-real gaps in train/test separation? More clarification in the paper would be helpful.


From the results, it seemed that the “path” instructions for languages, i.e. text instructions, does not provide much improvement gains compared to text goal. This is expected if we do not have enough diversity at the task and language level, but it’s also a bit surprising as the low-level language instructions seem to provide more details. I wonder about the specific details on this experiment, how to rephrase the languages, what encoder is used, and authors’ thoughts on why longer and more detailed instructions do not help much. Moreover, it seems that it’d be hard to provide language labels for simulation data, which often are similar task family and changing objects. I wonder how scaling is done on that direction (40x30 demos seem a lot in the real world). For example, figure 3’s left column seems to have mismatched language and images (it does not have the granola bar).


I am still trying to wrap my head around the stage 2, although it’s shown to be important in the experiments. I feel like this process can also be placed before stage 1 to make the encoder of “sparse information” better, and then apply the joint training? The way it’s put seems a bit ad-hoc. The policy (head) is not trained anymore in the second stage, and only the modality feature encoder are trained to match each other in this stage.


**Robotics Focus:**

Sufficient demonstration on hardware

**Summary Of Paper:**

This paper proposes a novel policy learning framework that allows flexible task specifications, which is an important problem for learning from different modalities. The method proposes a two-stage training method that provide supervision on cross-modality masking and distillations to enable better generalization performance. The method is evaluated on both simulation and real world experiments and demonstrated better performance compared to modality-specific training.

**Summary Of Recommendation:**

This paper presents a well-thought and clearly written method for solving policy learning with different modalitiy. The protocol and code (if released) are also very useful contribution for the community. There are still several weaknesses regarding experiments and algorithms and would love to hear about the response from the authors. Overall I recommend acceptance of this paper.

---

> ### Author Response · Authors · 2023-08-12
> **Continued Rebuttal (#2)**
>
> > I wonder how scaling is done on that direction (40x30 demos seem a lot in the real world).
>
> All the tasks used in the real world are mentioned in Figure 5 in the Appendix, and the details of collecting the dataset is added in Appendix Section 7.4. To summarize the dataset collection effort in the real world, the robot trajectories are collected using SpaceMouse by a human operator. The task specification dataset for each task is collected using ChatGPT followed by human correction for Text specifications, Amazon Polly for speech specifications, and using a human demonstration of the task (for visual specifications).
>
> > For example, figure 3’s left column seems to have mismatched language and images (it does not have the granola bar).
>
> We have fixed the error in Figure 3 and updated it in the updated version of the paper. Additionally, we provide more visualizations of dataset in the [website](https://sites.google.com/view/mutex/home#h.immf6bqvddja)
>
> **Why two stage training of Mutex in that order?**
>
> > The way it’s put seems a bit ad-hoc.
>
> Stage 1 of Mutex training uses masked modeling where each modality reinforces the other modalities to learn grounded, task-relevant information, where text and speech specifications help extract the correct information from visual specifications. On the other hand, visual specifications help in grounding text and speech specifications. In stage 2, we enrich other modalities with the more-informed video specification obtained after stage 1. The principle here is to propagate the information from the richer modality (also indicated by a better performance of its uni-modal model) into the others.
>
> >  I feel like this process can also be placed before stage 1 to make the encoder of “sparse information” better, and then apply the joint training
>
> Initially, the projection layer of video specification does not capture any useful information for the tasks (due to random initialization of weights), so applying stage 2 before stage 1 does not enrich information in other modalities.
>
> >  The policy (head) is not trained anymore in the second stage, and only the modality feature encoder are trained to match each other in this stage.
>
> This is a stimulating question, and we thank the reviewer for highlighting it. Stage 2 (i.e., cross-modal matching) only aims to enrich representations of task specifications while not losing robot action-relevant information. Jointly training the entire policy in stage 2 may result in a local optimum where video specification loses information to match other modalities' representation. To empirically verify our hypothesis, we run an empirical study (1) **Mutex (w/ finetuning full model**) by training the entire architecture end-to-end, (2) **Mutex (w/ finetuning video encoder**)  by fine-tuning the video encoder as well during cross-modal matching, and find that it results in a suboptimal solution in both scenarios corroborating our initial hypothesis that video representations may lose information. The results are summarized in the Table below.
>
> | Method                             | Text Goals | Text  Instructions | Image Goals | Video Demonstrations | Speech Goals | Speech Instructions |
> |------------------------------------|:----------:|:------------------:|:-----------:|:--------------------:|:------------:|:-------------------:|
> | Mutex (w/ finetuning full model)    |       37.1 |               40.2 |        44.7 |                 46.3 |         31.6 |                34.2 |
> | Mutex (w/ finetuning video encoder) |       49.6 |           **53.4** |        57.6 |                 61.3 |         38.4 |            **46.0** |
> | Mutex                               |   **50.9** |               53.0 |    **61.6** |             **63.2** |     **40.9** |            **46.0** |

---

### Official Review · Reviewer_JffC · 2023-07-17

**Confidence:** 4
**Originality:** Fair
**Technical Quality:** Fair
**Clarity Of Presentation:** Fair
**Impact:** 2

**Recommendation:**

Weak Accept: I recommend accepting the paper, but will not argue for my recommendation if the majority of other reviewers have a different opinion.

**Review:**

This is an ambitious effort to bring in multiple modalities of task description for both representation learning and control which is an important idea in Robotics. I also like the fact that it has been utilized in real world robotics experiments. However I have the following issues with this wok:

Technical: While the concept behind the work is important, it lacks sufficient supporting evidence in the form of experiments and studies to substantiate its claims and they are tested at the surface:
-- I have some issues with the main premise of the paper. It feels like the paper aims both perception and control which requires a lot more experiments and evidence to back these claims: which one of these is the man claim (it feels like both)?: (1) The unified representation is more rich compared to learning representations based on individual modalities. If this is the claim, then this work lacks ablation experiments and results that demonstrates for example masked training performance in each modality is improved when combined with the other modalities independent of the second stage (control); (2) If the claim is that the unified representation is superior for control, my primary concern is that this work fails to adequately address crucial control issues, such as the compounding error factor present in vanilla behavior cloning.

-- "novel architecture based on transformers": the architecture is not that novel, adding and moving around self-attention and cross attention blocks do not necessarily makes it novel. It is essentially a variant of transformer encoder-decoder architecture.

-- Lack of other baselines in experiments: the authors only compare to a baseline of using individual modalities. They could have at least compared it to approaches such as VIMA although different.

Paper organization and presentation:
(1) The paper's readability and coherence are compromised as it becomes challenging to maintain a consistent context throughout. The inclusion of plain text (e.g., sections 3.1 and 3.2) without accompanying figures, algorithmic view, or visualizations makes it more difficult to comprehend and follow the research.
(2) Additionally, there are a few readability and grammatical issues which need to be fixed.

**Quality Of The Limitations Section:**

Limitations are not well addressed

**Questions For Rebuttal:**

(1) "This loss is optimized in tandem with the policy learning objective (i.e., behavior cloning) so that the representation of the task specification also captures action-relevant information". Also the pathway exists for this through he combined losses, are there any more results to demonstrate this? For example, video modality for task description might be capturing this by large.

(2) Why not doing "voice to text, then only use text". I am wondering how much a raw audio signal would contribute to the unified representation. That is the part I mentioned in the review part where it lacks enough evidence or ablations in terms of representation learning.

(2) "MUTEX is not only more versatile thanks to its multi76 modal understanding, but it is also more performant, improving the robustness of the task execution on every single modality.", this is only partially true in a behavior cloning set up. I am not convinced the learned representation is more robust in face of compounding error effect.

(4) "Our goal is to learn a unified policy that performs a diverse set of tasks ...". It is more like a unified representation, I do not understand when it is called a unified policy.

(5) "The goal of each training task Ti is specified by k alternative task descriptions,". It is not clear how these alternatives are generated and monitored.

(6) "we recreate these conditions in our training procedure to obtain robust single-modality representations", where are the results showing this robustness?

Editorial suggestions:

(1) Remove the "..."in abstract.
(2) Need polishing of the terms and phrases such as: "the information denser one", ...
(3) Please : "we mask out intermediate regions of the features obtained from a pre-trained CLIP model", please describe what do you precisely mean by "intermediate regions of the features "

**Robotics Focus:**

Sufficient demonstration on hardware

**Summary Of Paper:**

This work presents an approach for learning a unified representation from a multimodal task representation and a collection of expert demonstrations for each task. The main premise of the this work is to allow cross-modal information to further enhance the representation required for learning robust policies to solve the downstream tasks from expert demonstrations. The approach, consists of two stages: (1) A mask modeling approach is used to allow for cross-modal information to be encoded in the learned representation; (2) A policy is learned using this representation to solve a task from any or a combination of task representation modalities. The paper then presents experimental results in both real world simulation and real world domains (100 tasks 14 in simulation and 40 tasks in the real world) and demonstrates improved performance when such representations are used.

**Summary Of Recommendation:**

I understand the technical material well however I do not have a broad knowledge of the relevant work except for a few. Independent of that, I think the claims are broad and the experiments do not provide enough evidence to back the claims. I recommend weak reject.

---

> ### Author Response · Authors · 2023-08-12
> **Continued Rebuttal (#2)**
>
> > -- Lack of other baselines in experiments: the authors only compare to a baseline of using individual modalities. They could have at least compared it to approaches such as VIMA although different.
>
> We thank the reviewer for the valuable suggestion. As the first reviewer mentioned, many of our experiments focus on the relative gain going from unimodal to multi-modal. There are no baselines that can handle such a multi-modal task specification as most prior works have focused on specifying tasks with only a single modality. The baselines presented in our work are representative and state-of-the-art modality-specific methods from the literature - [1,2,3] for language, [4,5] for image goal, [6,7,8] for video demonstration. In addition to baselines presented in the submission and based on the reviewers feedback, we trained uni-modal baselines using **R3M** [9] for Image Goals and Video Demonstrations, **T5** [10] for Text Goals and Text Instructions, and a bimodal **VIMA** [11] baseline, which uses both text goals and object images to specify tasks. The additional baseline results are summarized in the table below and Table 3 in the appendix of the attached updated paper. We see that Mutex consistently outperforms the uni-modal baselines and more recent bimodal VIMA demonstrating that training using our methodology for multimodal task specifications leads to higher performance than training with individual modalities. Details of the baselines are included in Appendix Section 7.5.
>
> |            Method            | Text Goals | Text  Instructions | Image Goals | Video Demonstrations | Speech Goals | Speech Instructions | Text Goals + Image Goals |
> |----------------------------|:----------:|:------------------:|:-----------:|:--------------------:|:------------:|:-------------------:|:------------------------:|
> |      T5 (w/ Text Goals)      |    40.0    |          -         |      -      |           -          |       -      |          -          |             -            |
> |   T5 (w/ Text Instructions)  |      -     |        44.0        |      -      |           -          |       -      |          -          |             -            |
> |     R3M (w/ Image Goals)     |      -     |          -         |     59.5    |           -          |       -      |          -          |             -            |
> | R3M (w/ Video Demonstration) |      -     |          -         |      -      |         44.7         |       -      |          -          |             -            |
> |             VIMA             |      -     |          -         |      -      |           -          |       -      |          -          |           47.0           |
> |             Mutex            |  **50.1**  |      **53.0**      |   **61.6**  |       **63.2**       |   **40.9**   |       **46.0**      |         **59.2**         |
>
>
> > The inclusion of plain text (e.g., sections 3.1 and 3.2) without accompanying figures, algorithmic view, or visualizations makes it more difficult to comprehend and follow the research.
>
> We thank the reviewer for the suggestions to help us improve the writing. We add pseudocode attached in Appendix Algorithm 1 in the updated version of the paper attached, and add reference in the main text [Ln144, Ln171].
>
> ## References
>
> [1] Jang, Eric, et al. "Bc-z: Zero-shot task generalization with robotic imitation learning.", 2022.
> [2] Brohan, Anthony, et al. "Rt-1: Robotics transformer for real-world control at scale.", 2022.
> [3] Nair, Suraj, et al. "Learning language-conditioned robot behavior from offline data and crowd-sourced annotation.", 2022.
> [4] Pathak, Deepak, et al. "Zero-shot visual imitation.", 2018.
> [5] Cui, Zichen Jeff, et al. "From play to policy: Conditional behavior generation from uncurated robot data.", 2022.
> [6] Yu, Tianhe, et al. "One-shot imitation from observing humans via domain-adaptive meta-learning.", 2018.
> [7] Bonardi, Alessandro, Stephen James, and Andrew J. Davison. "Learning one-shot imitation from humans without humans.", 2020.
> [8] Stephen et al. "Task-embedded control networks for few-shot imitation learning.", 2018.
> [9] Nair, Suraj, et al. "R3m: A universal visual representation for robot manipulation.", 2022.
> [10] Raffel, Colin, et al. "Exploring the limits of transfer learning with a unified text-to-text transformer.", 2020.
> [11] Jiang, Yunfan, et al. "Vima: General robot manipulation with multimodal prompts.", 2022.

---

> ### Author Response · Authors · 2023-08-12
> **Continued Rebuttal (#3)**
>
> ## Questions
>
> > (1) "This loss is optimized in tandem with the policy learning objective (i.e., behavior cloning) so that the representation of the task specification also captures action-relevant information". Also the pathway exists for this through he combined losses, are there any more results to demonstrate this? For example, video modality for task description might be capturing this by large.
>
> This is a great insight, and we also conducted experiments about this in the initial design of our approach. We found that the policy learning objective (BC loss) significantly impacts the learned representation. By incorporating the BC loss, we effectively captured the necessary information for robot action prediction.
>
>
> Although the video demonstration provides detailed information, we found that it alone is insufficient for capturing all the relevant information. Intuitively, learning the task specification representation without incorporating robot-action data may not fully capture the crucial information for action prediction. This is because none of the modalities contain information about the specific values of low-level robot actions, including the human video demonstration.
>
>
> To systematically evaluate this, we conducted an empirical study. We initially pretrain the task specification representations using the two-stage training process without utilizing the BC loss. Subsequently, we learned the policy using the frozen task specification representations. The results of this experiment are summarized in the table below. We see that jointly training with BC loss leads to better performance corroborating with our intuition that it helps capture action-relevant information.
>
>
> | Method                                 | Text Goals | Text  Instructions | Image Goals | Video Demonstrations | Speech Goals | Speech Instructions |
> |----------------------------------------|:----------:|:------------------:|:-----------:|:--------------------:|:------------:|:-------------------:|
> | Mutex (w/ Pretraining Representations) |    42.3    |        46.8        |      46     |          48          |     34.4     |         32.0        |
> | Mutex                                  |  **50.9**  |      **53.0**      |   **61.6**  |       **63.2**       |   **40.9**   |       **46.0**      |
>
>
> > (2) Why not doing "voice to text, then only use text".
>
> This is a great point; in theory, we could use voice-to-text and treat the transcribed speech modality as text. Is text easier to interpret than speech? Inspired by the reviewer's suggestion, we evaluate our Mutex model trained with six modalities on transcribed speech goals, treating them as text goals without additional training. Mutex performs better in **Transcribed Speech Goals (55.0%)** than **Speech Goals (40.9%)**. We hypothesize this is due to removing all variations in raw speech signals, including accent, tempo, and tone that make understanding speech harder than text. However, although less performant, raw speech offers additional information, like emotions and tempo, that can be used in future HRI research.
>
> >  I am wondering how much a raw audio signal would contribute to the unified representation.
>
> We thank the reviewer for suggesting an intriguing experiment. In addition to our prior experiment with a 6-modalities trained Mutex, we asked ourselves: is there a benefit of training with speech? To investigate the effect of raw speech signals on the performance of Mutex, we conduct an additional experiment where we train Mutex without speech descriptions, using text goals, text instructions, image goals, and video demonstrations. Maybe surprisingly, we found that the performance of Mutex trained with speech is superior to the model trained without speech [Summarized in the table below].This could be caused by the different statistics of text and speech specifications (speech is slightly more colloquial) providing additional signals to other modalities that help better understand the tasks specified. Nevertheless, speech provides a natural interface for humans to communicate the tasks, contains additional information such as emotion and tempo that can be relevant in HRI contexts, and shows to contribute positively in our experiments to the interpretation of other task specification modalities via cross-modal learning.
>
> | Method             | Text Goals | Text  Instructions | Image Goals | Video Demonstrations | Speech Goals | Speech Instructions |
> |--------------------|:----------:|:------------------:|:-----------:|:--------------------:|:------------:|:-------------------:|
> | Mutex (w/o Speech) |    43.0    |        49.2        |     59.2    |         60.1         |       -      |          -          |
> | Mutex              |  **50.1**  |      **53.0**      |   **61.6**  |       **63.2**       |   **40.9**   |       **46.0**      |

---

> ### Author Response · Authors · 2023-08-12
> **Continued Rebuttal (#4)**
>
> > (4) "Our goal is to learn a unified policy that performs a diverse set of tasks ...". It is more like a unified representation, I do not understand when it is called a unified policy.
>
> Major works have focused on learning a policy that can execute tasks specified by one single modality - language [1,2,3], image goal [4,5], and video demonstration [6,7,8], leading to siloed systems tailored to individual task specification modalities. In contrast, we propose a ‘unified’ policy that can execute tasks specified by any task specification modality, where unification is of modality-specific policies.
>
> > (5) "The goal of each training task Ti is specified by k alternative task descriptions,". It is not clear how these alternatives are generated and monitored.
>
> Details of generating the task specifications are mentioned in Appendix Section 7.4. To summarize, we use ChatGPT to generate specifications with enough variability for text goals and descriptions. For visual task specification modalities, a human achieves the same task with different instantiations. For speech specifications, we use Amazon Polly to generate speech with various commands and voice characters for training and testing. [Refer to [website](https://sites.google.com/view/mutex/home#h.immf6bqvddja) for examples]
>
> > (6) "we recreate these conditions in our training procedure to obtain robust single-modality representations", where are the results showing this robustness?
>
> In Table 1 and Table 2, we demonstrate that the representation of the task specifications learned using multiple modalities in Mutex leads to a more generalizable policy when subjected to unseen task descriptions. Quantitatively, our single model, Mutex, consistently outperforms all unimodal baselines with an average gain of **11% in simulation** and **8% in real-world**. We updated the paper to indicate the results in this sentence [Ln 152].
>
>
> ## Editorial Suggestions
>
> > (1) Remove the "..."in abstract. (2) Need polishing of the terms and phrases such as: "the information denser one", ...
> > -- "novel architecture based on transformers": the architecture is not that novel, adding and moving around self-attention and cross attention blocks do not necessarily makes it novel. It is essentially a variant of transformer encoder-decoder architecture.
>
> We thank the reviewer for the suggestions to help improve the writing of the paper, and have updated the abstract to incorporate the changes in the updated paper.
>
> > (3) Please : "we mask out intermediate regions of the features obtained from a pre-trained CLIP model", please describe what do you precisely mean by "intermediate regions of the features "
>
> Since reconstructing original pixel values is noisy, we mask and predict features obtained from the CLIP model instead. Specifically, for Image Goals, the features obtained after the 22nd block of the CLIP vision block are of dimensions [num_regions, embedding_dimension]. We mask and predict from these regional CLIP features extracted from the pretrained CLIP model. For video demonstration, we mask out individual frame features extracted from the CLIP model.
> The details of masking are added in Appendix Section 7.2. We add the reference in the main text to the appendix for more clarity.
>
> ### References
>
> [1] Jang, Eric, et al. "Bc-z: Zero-shot task generalization with robotic imitation learning.", 2022.
> [2] Brohan, Anthony, et al. "Rt-1: Robotics transformer for real-world control at scale.", 2022.
> [3] Nair, Suraj, et al. "Learning language-conditioned robot behavior from offline data and crowd-sourced annotation.", 2022.
> [4] Pathak, Deepak, et al. "Zero-shot visual imitation.", 2018.
> [5] Cui, Zichen Jeff, et al. "From play to policy: Conditional behavior generation from uncurated robot data.", 2022.
> [6] Yu, Tianhe, et al. "One-shot imitation from observing humans via domain-adaptive meta-learning.", 2018.
> [7] Bonardi, Alessandro, Stephen James, and Andrew J. Davison. "Learning one-shot imitation from humans without humans.", 2020.
> [8] Stephen et al. "Task-embedded control networks for few-shot imitation learning.", 2018.

---

> > ### Comment · Reviewer_JffC · 2023-08-14
> > **Need more experimentations for control**
> >
> > I am curious to learn whether this approach can create a more robust representation for control, better handling the compounding error effect that can arise in a behavior cloning scenario (initially discussed in S. Ross and D. Bagnell's work, "Efficient reductions for imitation learning"). While I find the approach promising, I'm not fully convinced that it has undergone comprehensive experimentation and analysis in the control domain. Vanilla behavior cloning may not reveal all the strengths and weaknesses of this approach when applied to control tasks.

---

> > > ### Author Response · Authors · 2023-08-15
> > > **Follow up response**
> > >
> > > Through our work (using behavior cloning to learn a policy), we demonstrate that Mutex learns a multimodal representation together with the policy that generalizes better to unseen task specifications and initial object locations compared to the traditionally used unimodal policies (**+11%** improvement in simulation, **+8%** in real-world). We firmly believe that our evaluation of Mutex using behavior cloning, including ablations and multiple recent baselines, is robustly and sufficiently evidencing the benefits of our approach. Here's why:
> > > 1. Behavior cloning (BC) has been proven to be data-efficient and applicable in multitask settings with hundreds of manipulation tasks, as demonstrated in previous studies [1,2,3].
> > > 2. Several seminal and recent representation learning papers have relied solely on BC as the control strategy [4, 5, 6, 7, 8].
> > >
> > > While it would be valuable to apply the general ideas of Mutex to other control strategies, such as reinforcement learning (like VC-1) or interactive IL, it is essential to note that testing the ideas presented in Mutex with a different control framework is a significant challenging because,
> > >
> > > 1. No other control-learning framework, to our best knowledge, has scaled to a multitask policy with hundreds of tasks in robot manipulation.
> > >
> > > 2. Mutex is intrinsically coupled with BC and is not a plug-and-play module [Refer [here](https://openreview.net/forum?id=PwqiqaaEzJ&noteId=-Z7vGWETcy)].
> > >
> > > ### References -
> > > [1] Shridhar, Mohit, et al. "Perceiver-actor: A multi-task transformer for robotic manipulation.", 2023.
> > > [2] Jang, Eric, et al. "Bc-z: Zero-shot task generalization with robotic imitation learning.", 2022.
> > > [3] Brohan, Anthony, et al. "Rt-1: Robotics transformer for real-world control at scale.", 2022.
> > > [4] Nair, Suraj, et al. "R3m: A universal visual representation for robot manipulation.", 2022.
> > > [5] Tony, Zhao. et al. “What Makes Representation Learning from Videos Hard for Control?”, 2023.
> > > [6] Parisi, Simone, et al. "The unsurprising effectiveness of pre-trained vision models for control.", 2022.
> > > [7] Lynch, Corey, et al. "Language conditioned imitation learning over unstructured data.", 2020.
> > > [8] Myers, Vivek, et al. "Goal Representations for Instruction Following: A Semi-Supervised Language Interface to Control.", 2023.

---

### Official Review · Reviewer_G6C8 · 2023-07-18

**Confidence:** 4
**Originality:** Very Good
**Technical Quality:** Very Good
**Clarity Of Presentation:** Good
**Impact:** 4

**Recommendation:**

Weak Accept: I recommend accepting the paper, but will not argue for my recommendation if the majority of other reviewers have a different opinion.

**Review:**

## Strengths

1. **Expanding beyond two modalities**: Most multimodal work focuses on images and text. By expanding out to video and audio and supporting instructions in *any* modality, the proposed method is much more flexible than most multimodal specifications. The authors devote a lot of effort to describing and exploiting the ways in which modalities can be complementary to each other.

2. **Hardware Evaluation**: The hardware evaluation is not just a proof-of-concept demonstration, and the paper includes quantitative results on how the method performed on the hardware.

3. **Dataset**: The real-world data the authors introduce will likely be useful for future work on multimodal task-specifications.

4. **Relevant Ablations**: The authors ablate out the relevant components of their model to support their claim that multimodal training improves performance on the tasks tested.



## Weaknesses
1. **Lack of baselines**: the model is evaluated against an ablated version of itself, but there are no external baselines. While this does not detract from the pitch of the paper as a multimodal training method (since the relative changes are important, not the absolute performance) it would be helpful to contextualize the results with other unimodal baselines, at least for the text goals/instructions and image goals, for which baseline systems exist.
Along the same lines, it wasn't clear to me how the unimodal baselines were trained.

2. **Lack of details**: There are a number of details which are omitted from the paper, without which I think reproducing the results would be difficult. These include hyperparameter settings like the masking ratio for different modalities, the loss mixture ratios, the specifications of the different encoders, the projection layers, policy encoder, decoder, etc.. This information would be fine to include as supplementary material, but should be provided with the paper somehow.

3. **Framing issues**: There is a bit of divergence between the framing of the paper and the experiments. One goal stated on L131-132 is to use multimodal training to generalize to novel tasks, but there are no experiments to this end.


## Comments
- The loss equation on L158-159 seems unneccessary, since it doesn't actually tell me what the loss between $y_i$ and $\hat{y}_i$ is. The equation also uses overloaded notation. All this equation tells me is that you averaged the loss across modalities, and you could save a lot of space by just writing that out.

- L226 has a typo or missing sentence in "...whereas a Kinect Azaure workspace camera."

**Quality Of The Limitations Section:**

Limitations are addressed clearly

**Questions For Rebuttal:**


# Questions
- L53 was very hard for me to parse because it uses "visual specification" twice in a row; what do you mean here by visual specification? Is that a blanket term for images and video?

- L183: why are gradients not back-propped into the video encoder? Is this because of memory considerations or performance considerations?


**Robotics Focus:**

Sufficient demonstration on hardware

**Summary Of Paper:**

This paper proposes a unified architecture for conditioning on task specifications in multiple modalities.
The modalities covered are video, audio, text, and images.
Broadly, the method involves learning a joint representation space for the multiple modalities that can be used for guiding manipulation tasks.
The method also suggests matching less information-dense modalities to video demonstrations, which are more information dense.
The paper also introduces a real-world dataset with multiple input modalities.
The method is evaluated in simulation and on hardware.
The authors find that multimodal pre-training improves their performance on instructions given in a single modality, and that both the multimodal pre-training and matching to video data result in performance increases across almost all modalities.

**Summary Of Recommendation:**

Based on the weakness section in my review, I'm recommending Weak Accept. I think the technical and data contributions of the paper are very strong, but I have qualms about the current presentation, especially about the lack of details which I think would hinder the reproducibility of the method.

---

> ### Author Response · Authors · 2023-08-12
> **Continued Rebuttal (#2)**
>
> > **Lack of details**: There are a number of details which are omitted from the paper, without which I think reproducing the results would be difficult.
>
> We are committed to publicly release the codebase and dataset used by Mutex to promote reproducibility, so all these values will be completely available.
> In addition, the architecture details are added in Appendix Section 7.3, and training hyperparameters are in Appendix Section 7.2 in the updated version of the paper attached.
>
> > **Framing issues**: There is a bit of divergence between the framing of the paper and the experiments. One goal stated on L131-132 is to use multimodal training to generalize to novel tasks, but there are no experiments to this end.
>
> In our work, we evaluate all the methods when subjected to both new initial conditions (i.e., the position of objects) and unseen task descriptions. We rephrase, in the updated version [Ln 129-131], to a more accurate -
> ```
>  We aim to create a policy that not only performs the $n$ tasks in the training dataset $D$ (seen tasks), even in new initial conditions (e.g. positions of the objects) but also generalizes to previously unseen task-descriptions.
> ```
>
> ## Questions
>
> > L53 was very hard for me to parse because it uses "visual specification" twice in a row; what do you mean here by visual specification? Is that a blanket term for images and video?
>
> Yes, visual specification is a general term, including both image and video specifications. Intuitively, by predicting the masked tokens, we promote inter-modal interactions, which enables the grounding of text and speech specifications while helping extract the right information from visual specifications. We changed that sentence to [Ln51-54]:
>
> ```
> Firstly, we exploit the complementary strengths of each modality, where text and speech specifications help in focusing task-relevant features from visual specification (image goals and video demonstration). On the other hand, visual specification helps to ground the text and speech through cross-modal interactions.
> ```
>
> > L183: why are gradients not back-propped into the video encoder? Is this because of memory considerations or performance considerations?
>
> We thank the reviewer for the insightful question. In cross-modal matching, we aim to enrich other modalities with video specifications containing the most information. We freeze the video encoder to avoid local optima during optimization in cross-modal matching, where video specification loses information. We empirically verified this and summarized the result in the Table below, where back-propagating gradients through video encoder lead to suboptimal performance.
>
> |                Method               | Text Goals | Text  Instructions | Image Goals | Video Demonstrations | Speech Goals | Speech Instructions |
> |-----------------------------------|:----------:|:------------------:|:-----------:|:--------------------:|:------------:|:-------------------:|
> | Mutex (w/ Finetuning Video Encoder) |    49.6    |      **53.4**      |     57.6    |         61.3         |     38.4     |       **46.0**      |
> |                Mutex                |  **50.1**  |        53.0        |   **61.6**  |       **63.2**       |   **40.9**   |       **46.0**      |
>
> ## Comments
>
> > The loss equation on L158-159 seems unneccessary, since it doesn't actually tell me what the loss between yi and y^i is. The equation also uses overloaded notation. All this equation tells me is that you averaged the loss across modalities, and you could save a lot of space by just writing that out.
>
> > L226 has a typo or missing sentence in "...whereas a Kinect Azaure workspace camera."
>
>  Thank you for the suggestions. We have made changes to the updated version of the paper Ln158 (removed), Ln224 to clarify these points.

---

> > ### Comment · Reviewer_G6C8 · 2023-08-15
> > **Reviewer response**
> >
> > Thanks for the thorough response addressing my concerns. I'm satisfied with the additional baseline results and the additional details, and recommend that the paper be accepted.

---

> > > ### Author Response · Authors · 2023-08-15
> > > **Thank you**
> > >
> > > We thank the reviewer for the insightful comments and suggestions.

---

### Author Response · Authors · 2023-08-16
**Message to AC**

Dear AC,

We would like to bring your attention to one of the reviews [by Reviewer JffC
] that we have received. We strongly believe that demonstrating our results using Behavior Cloning for policy learning provides sufficient evidence to prove the effectiveness of Mutex, given the baselines and ablations supplied in the paper and added during the rebuttal. Expecting to implement the ideas presented in our work with multiple policy learning frameworks is an unreasonable ask. We have given our reasoning [here](https://openreview.net/forum?id=PwqiqaaEzJ&noteId=XSQgiHuPwB). We hope you look into the review more critically and make the judgment.

Thank you.

---

### Decision · Program_Chairs · 2023-08-30

**Decision:**

Accept (Poster)

**Comment:**

The reviewers unanimously vote for accepting the paper and I concur. Some reviewers raised concerns about writing and presentation. G6C8 remarks, "but I have qualms about the current presentation". A closer look at the draft reveals that even the first sentence in the abstract might be incomplete. There are also concerns about reproducibility and comparisons to the baselines that authors presented in the rebuttal. I encourage the authors to address the presentation concerns and include the baseline comparisons in the final version of their paper.